# Deviations from Mendelian Inheritance on Bovine X-Chromosome Revealing Recombination, Sex-of-Offspring Effects and Fertility-Related Candidate Genes

**DOI:** 10.3390/genes13122322

**Published:** 2022-12-09

**Authors:** Samir Id-Lahoucine, Joaquim Casellas, Pablo A. S. Fonseca, Aroa Suárez-Vega, Flavio S. Schenkel, Angela Cánovas

**Affiliations:** 1Centre for Genetic Improvement of Livestock, Department of Animal Biosciences, University of Guelph, Guelph, ON N1G 2W1, Canada; 2Departament de Ciència Animal i dels Aliments, Universitat Autònoma de Barcelona, Bellaterra, 08193 Barcelona, Spain

**Keywords:** transmission ratio distortion, chromosome X, recombination, bovine

## Abstract

Transmission ratio distortion (TRD), or significant deviations from Mendelian inheritance, is a well-studied phenomenon on autosomal chromosomes, but has not yet received attention on sex chromosomes. TRD was analyzed on 3832 heterosomal single nucleotide polymorphisms (SNPs) and 400 pseudoautosomal SNPs spanning the length of the X-chromosome using 436,651 genotyped Holstein cattle. On the pseudoautosomal region, an opposite sire-TRD pattern between male and female offspring was identified for 149 SNPs. This finding revealed unique SNPs linked to a specific-sex (Y- or X-) chromosome and describes the accumulation of recombination events across the pseudoautosomal region. On the heterosomal region, 13 SNPs and 69 haplotype windows were identified with dam-TRD. Functional analyses for TRD regions highlighted relevant biological functions responsible to regulate spermatogenesis, development of Sertoli cells, homeostasis of endometrium tissue and embryonic development. This study uncovered the prevalence of different TRD patterns across both heterosomal and pseudoautosomal regions of the X-chromosome and revealed functional candidate genes for bovine reproduction.

## 1. Introduction

Significant deviations from Mendelian inheritance, well known as transmission ratio distortion (TRD), have been observed in a number of diploid organisms [1,2,3,4,5]. It is widely recognized that the TRD phenomenon is linked to a broad range of biological mechanisms associated with basic processes of life that underlie meiosis, gametogenesis, sperm and ova fertility and viability at developmental stages of the reproductive cycle (e.g., embryos, postnatal metabolism and growth, etc.) [6,7]. The importance of the above-mentioned events coupled with the current widespread use of DNA technologies, such as genotyping of breeding stock on a routine basis in some livestock industries (e.g., beef and dairy cattle and swine), emphasizes the relevance and potential usefulness of assessing TRD and opening new opportunities for research and extension initiatives. Although TRD remains an unclear and ambiguous phenomenon [8,9], it can be viewed as the ultimate consequence of several genetic factors arising at different stages of the reproductive process and early neonatal life. Indeed, some cases of TRD, such as the absence of homozygosity, are already used to target recessive phenotypes that affect reproduction, for example, decreased fertility and embryo loss in different livestock species [5,10,11,12,13].

Despite the recent interest in targeting TRD regions across the whole genome or, alternatively, evaluating for the absence of homozygous haplotypes in livestock species [4,5,10,11,13,14,15], TRD on sex chromosomes has not yet received attention. In fact, sex chromosomes have been discarded in a wide range of genetic studies of livestock species due to their large heterosomal (i.e., non-pseudoautosomal) part, which requires adaptation of standard analytical approaches. Nevertheless, two studies in humans have found a specific sex-offspring TRD on the X chromosome [16,17].

In this study, we aimed to investigate the inheritance of single nucleotide polymorphisms (SNPs) and haplotypes from parents to offspring in order to characterize and understand the biology of TRD for the bovine sex chromosomes. The main objectives of this research were: (1) to generalize TRD models with male- and female-offspring-specific TRD components, (2) to determine the pseudoautosomal and heterosomal parts of the sex chromosomes using trios (sire-dam-offspring genotypes), (3) to characterize TRD on both heterosomal and pseudoautosomal parts of the Holstein cattle sex chromosomes, and finally, (4) to annotate genes and to evaluate the functional consequences of TRD regions via functional analyses.

## 2. Materials and Methods

### 2.1. Genotype Data

Data used in this study comprised a total of 436,651 Holstein dairy cattle genotypes provided by the Canadian Dairy Network, a member of Lactanet (Guelph, ON, Canada). These animals were sampled from all available Holstein genotypes (>1 million genotypes; October 2017) and were genotyped within the first 90 days after birth to minimize offspring selection issues [13,18]. Animals were genotyped by various SNP genotyping arrays ranging from 2900 to 777,962 SNPs (Appendix A) and mapped to the UMD3.1 *Bos taurus* genome assembly. All SNP markers mapped in sex chromosomes and genotyped for at least 10 sire-dam-offspring trios were selected for analyses, resulting in a total of 6577 SNPs distributed across the X-chromosome (Appendix A). The number of trios available for SNPs ranged from 10 to 340,332 with an average of 18,980 trios. The maximum number of genotyped sires and dams was 5917 and 130,501, respectively.

### 2.2. Imputation

In order to carry out haplotype analyses, 974 SNPs were selected from the heterosomal part of the X-chromosome (Appendix A). The number of raw genotypes ranged from 23,704 to 402,267 with an average of 141,821 genotypes across 974 SNPs. The length of the X-chromosome covered by these markers was between 29,245 and 137,012,896 bp. Data was phased and imputed using a Fortran program using familial information (i.e., parent-offspring trios and sibling pairs) with variable sliding windows ranging from 2 to 200 SNPs.

### 2.3. Male- and Female-Offspring Specific-TRD

Taking an autosomal (or pseudo-autosomal) chromosome region as a starting point, the analysis of TRD for male (_♂_) and female offspring (_♀_) can be carried out by modeling the probability of transmission of alleles from heterozygous parents [4,14]. Considering male offspring as an example, the probability of transmission of the A allele from heterozygous parents (A/B) can be stated as:P(A) = 1 − P(B) = 0.5 + α_♂_ and P(B) = 1 − P(A) = 0.5 − α_♂_,
where: α_♂_ is the TRD parameter ranging between −0.5 and 0.5. As described by Casellas et al. (2017) [4], this model can be easily expanded to account for sire- (α_s♂_) and dam-specific (α_d♂_) TRD. The same applies to female offspring to obtain α_♀_, α_s♀_ and α_d♀_.

The previous models can be generalized to the analysis of TRD phenomena in the non-autosomal (i.e., heterosomal) region of the X chromosome, but restricted to dam-specific TRD.

### 2.4. Statistical Analyses

Two different approaches were used for analyzing TRD across the bovine X-chromosome. Transmission ratio distortion was evaluated using both SNP-by-SNP (a total of 6577 SNPs) and a sliding windows approach of 2-, 4-, 10- and 20-SNP across 974 imputed SNPs as described by Id-Lahoucine et al. (2019) [13]. For sliding windows, the biallelic-haplotype procedure was applied. All the analyses were performed within a Bayesian framework using an adapted version of TRDscan software [13] with a unique Monte Carlo Markov chain of 110,000 iterations, where the first 10,000 iterations were discarded as burn-in. The statistical significance of the TRD was tested using a BF [19].

### 2.5. Pseudoautosomal Characterization

Mendelian inconsistency was estimated under a pseudoautosomal and heterosomal inheritance models using trios of genotypes (from dam, sire and offspring) in order to determine the nature of SNPs across the X-chromosome. Both analyses of LD and recombination events were performed. For recombination events, the sum of recombination probabilities was calculated for markers in pseudoautosomal parts of the X-chromosome using the package hsphase in R [20]. In addition, for better characterization of the pseudoautosomal regions on X- and Y- chromosomes, the regions with pseudoautosomal inheritance models were initially selected to perform a “nucleotide BLAST” analysis on Basic Local Alignment Search Tool (BLAST; https://blast.ncbi.nlm.nih.gov/Blast.cgi accessed on 10 December 2020) against the bovine reference genome. The sequence was retrieved using the bovine UMD3.1 genome assembly from NCBI database. After this step, a short sequence of 20 bp around the genomic position of the markers was extracted to obtain the nucleotide context around the TRD markers. Using the genome data viewer, regions with homology between the short sequence around the markers and the Y-chromosome were investigated. The whole assembly of the bovine Y-chromosome is completely available for visualization only for the bovine genome assembly Btau_4.6.1 and, therefore, this assembly was used for the data visualization of the Y-chromosome. It is important to reinforce that neither for UMD 3.1 or ARS-UCD 1.2 an assemble of bovine Y-chromosome is available, which results in the impossibility to use these assemblies for the abovementioned analysis.

### 2.6. Functional Analyses

The genes mapped around the TRD regions identified by both SNP- and haplotype- based methods were annotated using R software (Version 3.2.3) and the R package: Genomic functional Annotation in Livestock for positional candidate Loci (GALLO) [21]. The genes mapped around the TRD regions identified using the SNP approach were annotated using an interval of 100 Kilobase (Kb) upstream and downstream from the SNP position. For the haplotype approach, the genes were annotated using the start and end coordinates from each haplotype based on the UMD 3.1 assembly from the bovine reference genome. Subsequently, in addition to the bovine Ensembl ID and official associated Gene Symbol, the respective orthologous genes in human and mouse were also annotated for each positional candidate (Ensembl ID and Gene Symbol) using the R package biomaRt [22]. In this step, only the human ortholog showing a similarity higher than 75% with the bovine annotated gene were retained for subsequent analyses. The highest |α| and the respective log_10_(BF) of the associated marker (SNPs or haplotypes with TRD) were assigned for each positional candidate gene.

Functional analysis were performed using the IPA (Ingenuity Pathways Analysis) software (QIAGEN Redwood City, www.qiagen.com/ingenuity 15 December 2020) to identify the canonical metabolic pathways, diseases and functions associated with the list of candidate genes harboring deviations from Mendelian inheritance on the bovine X-chromosome (including both SNPs and haplotypes). In addition, functional analysis was performed to explore the existence of signaling networks connecting those genes [23]. The significantly enriched metabolic pathways were identified using a significance threshold of false-discovery rate (FDR) < 0.05.

## 3. Results and Discussion

### 3.1. Characterization of X-Chromosome Using Parent-Offspring Genotyped Trios

The mammalian sex chromosomes are characterized by two main types of regions: a short region of sequence homology between the X- and the Y-chromosomes, known as the pseudoautosomal region and a longer heterosomal (sex-linked) region specific for each sex chromosome. The pseudoautosomal region in cattle is located in the long arm of the X-chromosome and the short arm of the Y-chromosome. Its physical length is estimated to be between 5–9 Mb [24]. Other studies specified a size of 8 Mbp [25] and 5.7 Mb on the ARS-UCD1.2 reference genome assembly [26]. Within the pseudoautosomal region, pairing and recombination occurs between X- and Y-chromosomes in males as it does for autosomal chromosomes. On the other hand, the heterosomal regions do not pair and recombine during meiosis in males, due to the reduced homology between the chromosomes. In this study, the patterns of Mendelian incompatibilities, estimated under pseudoautosomal and heterosomal inheritance models for each individual SNP together with the patterns of their adjacent SNPs, were used to confirm their nature (i.e., pseudoautosomal or heterosomal). The short arm of the X-chromosome and part of the long arm showed high incompatibility under the pseudoautosomal inheritance model whereas null Mendelian inconsistencies under the heterosomal inheritance model (29,245–137,034,696 bp; Figure 1). Alternatively, the extreme part of the long arm of the X-chromosome (143,865,210–148,816,634 bp) displayed high and null Mendelian inconsistencies under heterosomal and pseudoautosomal inheritance models, respectively. This opposite behavior supported its pseudoautosomal nature described in the bovine genome [24,25,27]. However, unique individual SNPs showed Mendelian incompatibilities discrepant with their adjacent SNPs, suggesting possible genotyping artifacts or inaccurate annotation. These discrepant individual SNPs were removed from further TRD analyses. On the other hand, small regions of adjacent SNPs between these two clear extremes (heterosomal and pseudoautosomal parts) fitted either a heterosomal or a pseudoautosomal inheritance model.

From the preliminary scan, 5935 and 780 SNPs were shown as exhibiting heterosomal and pseudoautosomal inheritance, respectively. In order to guarantee a reasonable statistical power, only SNPs with at least 100 trios (sire-dam-offspring) of genotypes and a Mendelian inconsistency < 1% were maintained for TRD analyses (Appendix A). After this editing, the number of SNPs was reduced to 3832 and 400 with heterosomal and pseudoautosomal nature, respectively. The two main parts of the X-chromosome included 3680 heterosomal SNPs (BTX:29,245–137,034,696 bp) and 322 pseudoautosomal SNPs (BTX:143,865,210–148,816,634 bp). The undetermined regions of the X-chromosome (BTX:137,034,696–143,865,210 bp) displayed different inheritance patterns in the following order: 8 pseudoautosomal SNPs (BTX:137,181,696–137,476,239 bp), 9 heterosomal SNPs (BTX:137,531,272–137,717,498 bp), 54 pseudoautosomal SNPs (BTX:137,951,394–139,042,034 bp), 30 heterosomal SNPs (BTX:139,126,989–140,104,202 bp), 16 pseudoautosomal SNPs (BTX:140,116,340–140,380,961 bp) and 113 heterosomal SNPs (BTX:140,445,475–143,832,372 bp). The total sum of these parts was 141.6 Mb and 6.6 Mb for pseudoautosomal and heterosomal regions, respectively. The physical length of pseudoautosomal regions is in accordance with Das et al. (2009) [24], i.e., between 5–9 Mb. It must be noted that the pseudoautosomal part of the cattle X-chromosome is demarcated by the pseudoautosomal boundary, a border where the sequence similarity between the X- and the Y-chromosomes decreases (to 80–50%) and regions specific to each sex chromosome begin [28].

Couldrey et al. (2016) [25], reported 2 regions (presumably pseudoautosomal) with male heterozygosity greater than 1%, extending from 137 to 141 Mb and from 144 Mb to the end of the chromosome, providing evidence of the different inheritance patterns observed between 137 and 144 Mb. In addition, McClure et al. (2018) [29] characterized 101 pseudoautosomal SNPs for the X-chromosome from 137,330,600 to 148,482,298 bp. Among these 101 pseudoautosomal SNPs, 69 were in accordance with our results based on Mendelian compatibilities. However, 10 SNPs were assumed heterosomal in nature according to our results whereas were pseudoautosomal by McClure et al. (2018) [29]. Specifically, these 10 SNPs displayed ≥ 9% and ≤0.31% Mendelian inconsistencies under pseudoautosomal and heterosomal inheritance models, respectively, and their heterosomal nature was supported by the similar pattern of the corresponding adjacent SNPs. These SNPs are also reported with null male heterozygosity (≤0.02) and moderate-to-high female heterozygosity (≥0.29) by McClure et al. (2018) [29].

### 3.2. Deviations from Mendelian Inheritance on the Pseudoautosomal Part of the Sex Chromosomes

Decisive evidence with Bayes factor (BF; ≥100) according to Jeffreys’ scale [30] was identified for TRD across the Holstein sex chromosomes. The pseudoautosomal SNPs on the sex chromosomes revealed an important pattern of sire-TRD with specific sex-offspring components. A total of 180 SNPs was characterized by significantly displaying opposite sire-TRD between offspring sexes (Figure 2A; Appendix A). SNPs that exhibited specific sire-TRD via one single offspring sex (only male or female) were also observed, being 26 SNPs with sire-TRD via male and 19 SNPs with sire-TRD via female (Table 1; Appendix A). In addition, 3 SNPs displayed an opposite sire-TRD between offspring sexes and also a significant dam-TRD pattern only via one single offspring sex (Appendix A). On the other hand, additional 5, 2 and 1 SNPs were also identified with BF ≥ 100 for overall TRD (specific offspring sex), dam-TRD and sire-TRD (unspecific offspring sex), respectively. Among them, only one individual SNP showed a probability of random TRD ≤ 0.001 (BTX:147,104,359; sire-TRD = −0.04 with BF = 10^20.18^, 625 heterozygous sires and 1458 under-represented offspring).

### 3.3. Opposite Sire-TRD between Male- and Female-Offspring

Opposite sire-TRD between male- and female-offspring were observed across the whole pseudoautosomal part of sex chromosomes with a specific pattern (Figure 2). This TRD pattern was observed with a preferential transmission of one specific allele to male-offspring and the opposite allele to the female-offspring. Figure 3 shows how strong the sire-TRD was between male- and female-offspring among different matings for the SNP (BTX:144,617,673) with the highest BF. Taking into account the approximate empirical null distribution of TRD [13] at ≤0.001% margin error for random TRD, the number of regions with this pattern reduced from 180 to 149 SNPs spanning the length of the pseudoautosomal part of the sex chromosome. Here, the important discovery resides, as shown in Figure 2A, in the strong sire-TRD observed for SNPs at the beginning of the pseudoautosomal region and its marginal reduction with the distance until being null on the extreme of the X-chromosome. It is important to highlight that this pattern is primarily attributed to the main pseudoautosomal part of the chromosomes (i.e., 143,865,210–148,816,634 bp), where the first 8 SNPs (143,865,210–144,085,178) displayed TRD ranging from 0.47 to 0.50 and with full linkage to one sex (i.e., null recombination). This result indicates a strong linkage of some alleles to X- or Y-chromosomes at the beginning of pseudoautosomal region. Addition, the decay of opposite sire-TRD between male- and female-offspring with the distance describe the accumulated recombination events across the pseudoautosomal part of the cattle genome and suggest a consistent level of these events. The results of recombination analyses supported these patterns showing an absence of recombination events at the beginning of this pseudoautosomal genomic region and a more consistent level of recombination events started as TRD magnitude decreased (Figure 4). Therefore, specific sex-offspring TRD revealed another way to study recombination and linkage disequilibrium (LD) in sex chromosomes. Thus, whereas marker alleles in strong LD to specific sex chromosome (X or Y) were observed at the beginning of the pseudoautosomal region, this LD decreases at longer distances from the beginning of the pseudoautosomal region until being null on the extreme of the X-chromosome. This expected partial sex linkage on pseudoautosomal regions was previously reported in humans [31]. Moreover, this result suggested that novel mutations generated at the beginning of the pseudoautosomal part will be linked to a specific sex for long time than potential mutation occurred on the extreme of the X-chromosome.

This opposite sire-TRD pattern could reinforce the high recombination rate in the cattle pseudoautosomal region, as it was also described on other mammalian species, which exceed the genome average by 10–20 times [32]. Notice that the recombination events on the sex chromosomes in males are expected to be extraordinarily restricted to a short genomic segment (i.e., pseudoautosomal regions), appearing to be a recombinational hotspot in male meiosis [31]. In addition, the crossover in the pseudoautosomal region was described to be essential for the proper disjunction of X- and Y-chromosomes in male meiosis and a deletion of pseudoautosomal region was reported to result in male sterility [33]. However, biologically, the mechanisms by which this mandatory crossover is achieved remains unknown [33]. Moreover, the results of estimated LD using squared correlation between alleles also supported the high recombination rate in the cattle pseudoautosomal region. A decay of LD in the main pseudoautosomal region in comparison to the heterosomal and autosomal regions was observed (Appendix A). Furthermore, a relatively small decay of LD on heterosomal region in relation to autosomal region was also observed (Appendix A).

On the other hand, 15 pseudoautosomal SNPs (from 180) with opposite sire-TRD patterns were in the undetermined regions (i.e., BTX:137,181,696–137,476,239 bp, BTX:137,951,394–139,042,034 bp and BTX:140,116,340–140,380,961 bp). In contrast to the main pseudoautosomal part, we did not observe a similar pattern (i.e., the decreased magnitude of TRD with distance). It is important to emphasize that opposite sire-TRD between male- and female-offspring not necessarily imply recombination between X- and Y-chromosomes. Therefore, this pattern could also be explained by different linkage of alleles to X- and Y-chromosome.

### 3.4. Specific Sex-Offspring Sire-TRD

Across the X-chromosome, we identified 45 initial SNPs that are exhibiting sire-TRD via one single sex whereas null TRD in the opposite sex. After discarding possible random TRD at ≤0.001%, according to the approximate empirical null distribution of TRD [13], 8 and 7 SNPs showed male- and female-offspring sire-TRD, respectively (Table 1). As introduced previously, similar patterns of TRD on only one sex (specific sex-offspring TRD) were already identified in the human X-chromosome using microsatellite loci [16]. In fact, related patterns were also described for some disease (e.g., retinoblastoma) and were hypothesized to be associated with defective imprinting [34]. In addition, the sire-TRD pattern could be due to the fact that DNA sequences remain methylated for a much longer period in the male germline than in the female germline [35]. Within this context, an adequate imprinting may be required for viability of male or female embryos and the identified SNPs could be mapped in potential genes with differential imprinting patterns. It is notably important to mention that some of these SNPs may be in the same genes as distance between most of them was small (9 SNPs were at < 500 kb from the adjacent significant SNP).

The different magnitudes of TRD could also support imprinting for some of them, as TRD ranged from 0.04 to 0.36 in male-offspring and between 0.03 to 0.29 in female-offspring. For these SNPs, the number of under-represented offspring ranges from 47 to 1014 (1 SNP with ≥1000) for males and from 54 to 2183 (3 SNP with ≥1000) for females. The number of heterozygous sires ranges from 7 to 302 and from 13 to 1281 for male- and female-offspring, respectively. The minor allele frequencies of the detected SNPs ranges from 0.08 to 0.48. It is important to mention that moderate-to-high TRD magnitude signals were observed in regions with low or extremely low frequencies, suggesting the effects of TRD selection as observed in autosomal chromosomes as well [13]. Notably, the moderate frequencies of some SNPs in this case could be explained by the null TRD observed in one parent and/or offspring (male or female). Nevertheless, mechanisms such as imprinting could have the same consequences since the not preferentially transmitted allele is still observable in the offspring generation.

### 3.5. Deviations from Mendelian Inheritance on the Heterosomal Part of the X-Chromosome

#### 3.5.1. SNP-by-SNP Analysis

Among 3832 heterosomal SNPs, 12 SNPs were significantly detected with decisive evidence (BF ≥ 100) for dam-TRD (Table 2). Among them, 7 SNPs were significantly observed via male, 2 via female and 3 via both (i.e., non-sex-offspring TRD). However, these SNPs were reduced to 9, 8 and 2 considering a chance of random TRD of ≤0.1, ≤0.01 and ≤0.001%, respectively. The SNP with the highest BF displayed 3675 under-represented offspring with α_d_ = 0.02. The maximum magnitude observed for dam-TRD was |0.12| and displaying 287 under-represented offspring. In general, the prevalence of TRD across the heterosomal regions was small in comparison to pseudoautosomal and autosomal regions. This observation could be related to the biological differences between sexes, such as the processes of germ cell formation in males versus females (e.g., cell divisions rates in spermatogenesis versus oogenesis) [36].

#### 3.5.2. Haplotype Analysis

Across 974 SNPs of the heterosomal part of the X-chromosome, haplotypes displaying TRD were detected with decisive evidence of BF (≥100). After discarding regions with <10 heterozygous dams and <50 informative offspring, the number of regions was 944, being 44, 71, 282 and 597 detected for 2-, 4-, 10- and 20-SNP haplotypes, respectively. The number of regions at <0.001% margin of error reduced to 14, 26, 138 and 319, respectively. When correcting for multiple tests with kernel smoothing, the number of regions reduced to 69 haplotype windows explaining the observed TRD among the corresponding linked regions. The TRD was observed in both male- and female-offspring and with strong magnitudes ranging up to |0.48|. The top 20 regions according to BF are summarized in Table 3 and the complete list in the Appendix A. It must be emphasized that, even though 69 haplotypes were characterized, the maximum number of under-represented offspring observed was 193, showing a reduced relevance in comparison to autosomal chromosomes.

### 3.6. Recessive TRD Pattern

Recessive TRD pattern (i.e., lethality only in homozygous state) was targeted on pseudoautosomal SNPs (in both male- and female-offspring) and heterosomal SNPs/haplotypes (only female-offspring) on sex chromosomes. Whereas no recessive TRD patterns were found in SNP-by-SNP analyses, haplotype analyses revealed eight haplotypes (Table 4, Appendix A). These regions were detected with the genotypic TRD model [3,37] exhibiting additive-TRD ranging from −0.34 to −0.56. The unfavorable effects of the recessive alleles were compensated by positive dominance-TRD effects (ranging from 0.21 to 0.28) in heterozygous offspring, allowing for their survival. The corresponding number of non-observed homozygous female-offspring was between 5 to 8 (Table 4). In comparation to autosomal chromosomes, Holstein haplotype 3 was initially identified by 7 non-observed homozygous offspring from heterozygous carrier sire by heterozygous carrier maternal grandsire matings [10].

### 3.7. Functional Impact around the Pseudoautosomal Boundaries Identified Using Opposite TRD Patterns between Male and Female Offspring

Among the SNPs with highest opposite sire-TRD between male- and female-offspring, the 20 bp sequence around BTX:143,928,317 SNP was identified within the *LOC790278* pseudogene on the Y-chromosome, which has a corresponding version on the X-chromosome (*SHROOM2* gene). The pseudogene *LOC790278* is mapped on the heterosomal region of Y-chromosome (based on the Btau 4.6.1 annotation), before the SINE element responsible to reduce the homology between the X- and Y-chromosomes in the boundary between the heterosomal and pseudoautosomal [38]. Additionally, the region with the highest homology between the X-chromosome region and the Y-chromosome used in the BLAST analysis is also located in the predicted heterosomal part of the Y-chromosome. This region comprised 148.3 Kb (BTX:144,251,086–144,399,468) and has more than 99% similarity with the Y-chromosome (Appendix A). It is important to highlight that, as shown in Figure 2, even the sequential structure (regarding the gene order) of the homologous region and the BTX:143,928,317 SNP remains in the predicted heterosomal region of Y-chromosome. Within this region of homology between the pseudoautosomal region of the X-chromosome and predicted heterosomal region of the Y-chromosome, no TRD was detected due to the absence of markers exactly in this genome range, as shown in Figure 2C. As described by Raudsepp et al. (2012) [27], the *SHROOM2* is a X-specific gene in cattle. However, following the gene annotation from NCBI genome data viewer, this gene is mapped to the boundary between the pseudoautosomal and the heterosomal regions, due to the location of *GPR143*, the gene responsible to define the beginning of the pseudoautosomal region [38]. The results obtained here indicate a possible miss-assembly of the pseudoautosomal region of Y chromosome in Btau 4.6.1 or a complex rearrangement of the pseudoautosomal Y region, which must be better studied due to the functional importance of these regions. Additionally, the functional interpretation of the candidate markers in these regions can be affected since the causal variant could be mapped to both chromosomes.

### 3.8. Enriched Metabolic Pathways, Diseases and Biological Functions Associated with the Positional Candidate Genes Displaying Deviations from Mendelian Inheritance on the X-Chromosome

Functional annotation of positional candidate genes displaying deviations from Mendelian inheritance on the bovine X-chromosome was performed for the 55 detected regions: 15 SNPs with specific sex-offspring sire TRD (pseudoautosomal region), 12 SNPs with dam-TRD, 20 haplotypes with dam-TRD and 8 haplotypes with recessive TRD (heterosomal region). A total of 408 genes were annotated in an interval of 200 Kb (100 Kb up- and downstream) for the individual SNPs or within the haplotype coordinates. A total of 113 genes were annotated in the interval harboring the individual SNPs (among them, 60 genes had TRD signals within the gene coordinates) and 296 within the haplotype coordinates. It is important to highlight that some genes were annotated in both SNP and haplotype approaches. The human and mouse orthologs were used for those genes without an assigned gene symbol in the bovine reference genome. A total of 295 genes on TRD regions (out of the initial 408) were annotated. This list of 295 genes was used to perform the functional analysis using the IPA (Ingenuity Pathways Analysis) software to identify the enriched canonical metabolic pathways (Appendix A), diseases, and biological functions (Appendix A). The top 10 metabolic pathways, diseases, and functions identified are presented in Table 5. Despite no metabolic pathways were significant after adjustment for multiple testing (FDR < 0.05), the following canonical metabolic pathways were in the list of the top 10 enriched pathways at *p*-values < 0.05: citrulline degradation, spermine biosynthesis, phosphoribosyl diphosphate biosynthesis I, arginine biosynthesis IV, proline biosynthesis II (from arginine), urea cycle, pentose phosphate pathway (non-oxidative branch), glycine cleavage complex, phosphatidylcholine biosynthesis I, and acetyl-CoA biosynthesis I (pyruvate dehydrogenase complex). Among these metabolic pathways, the spermine biosynthesis plays a crucial role in male fertility. The gene associated with this pathway, in the current study, was the *SMS* (Spermine Synthetase) gene. Deletions in the X-chromosomal loci harboring the mice orthologous (*SpmS*) are associated with several impaired development phenotypes, such as sterility and profound postnatal growth retardation [39]. Another enriched pathway was the phosphatidylcholine biosynthesis, which was associated with the *PCYT1B* gene in this study, which had a haplotype exhibiting TRD. The disruption of function of the first enzyme in the phosphatidylcholine biosynthesis, the choline kinase α, causes embryonic lethality in mice [40]. Female knockout mice for the *PCYT1B* gene exhibited ovarian tissue disorganization with progressive loss of follicular formation and oocyte maturation [41].

Using the list of 295 positional candidate genes, 500 diseases and biological functions were enriched (FDR < 0.05). Among the top 10 enriched diseases and biological functions identified, a strong association between the genes and X-linked syndromic phenotypes was observed. It is important to highlight that several X-linked syndromes caused by structural alterations (copy number variations, translocations, inversions, breakpoints, etc.; Table 5) display a combination of neurological and reproductive alterations, such as X-fragile, Turner, Klinefelter, Kallmann, and Rett syndromes [42,43,44].

In addition, among the other significant enriched diseases and biological functions, the enrichment analysis indicated that the candidate genes are associated with relevant biological processes associated with embryo and cellular development, such as cell viability, angiogenesis, development of cardiovascular tissue, migration of endothelial cells, growth of connective tissue, abnormal morphology of body cavity and neuritogenesis (Figure 5). The genes associated with these processes are mapped in regions where different TRD patterns were identified, all in heterosomal regions. The relationship between the TRD markers and the positional candidate genes is shown in Appendix A. Among these positional and functional candidate genes, only *PLP1*, *DRP2* and *BTK* did not show a direct association with embryonic development or reproductive processes base on the literature review performed for each gene.

### 3.9. Functional Candidate Genes Identified in the Candidate Regions with Deviations from Mendelian Expectations

The genes associated with regions with dam-TRD were: *RNF128*, *PIGA*, *PLP1*, *BMX*, *KDM6A*, *CSNK1A1*, *VEGFD*, *DRP2* and *bta-mir-221*. The *RNF128* gene was identified as differentially expressed [45] between high and low fertility heifers during mid-luteal phase of the estrous cycle. Mutations in the *PIGA* gene are associated with nocturnal hemoglobinuria [46]. Additionally, high rates of early embryonic lethality and low chimerism (in the surviving animals) are observed in mice knockedout for *PIGA* [47]. The *BMX* gene has a crucial role during the endothelial cell migration and angiogenesis [48] acting as a downstream *RAP1* effector in *VEGF*-induced endothelial cell activation [49]. An insertion/deletion polymorphism in *KDM6A* is significantly associated with litter size and growth traits in goats, indicating a possible association between this gene and the survival ratio during embryonic development and post-natal development [50]. The *CSNK1A1* acts as a regulator of DNA damage response in embryonic stem cells [51]. Additionally, the expression of *CSNK1A1* is increased by the injection of *miR-320* in mice oocytes [52]. Interestingly, knockout mice for *mmu-miR-320* show a reduced proportion of oocytes that develop into two-cell and blastocyst stage embryos [52]. The *VEGFD* gene has a critical role during embryonic vasculogenesis and angiogenesis in zebrafish, reinforced by the microinjection of *zVEGFD* mRNA into one-cell-stage embryos, which results in severe misguidance of intersegmental vessels and abnormal connection between dorsal aorta and caudal vein [53]. It is important to highlight that the previously described gene, *BMX*, acts in the same *VEGF*-induced endothelial cell activation pathway. In addition, regarding the *VEGF*-induced endothelial cell activation pathway, the zebrafish ortholog of *bta-mir-221* is required for the tip cell proliferation and migration in mosaic blood vessels during angiogenesis, a process which is regulated by *VEGF* [54].

Among the genes corresponding to male- and female-dam-TRD regions, 3 genes were highlighted: *FLNA*, *FGF13* and *MECP2*. Mutations in *FLNA gene* cause total gene disruption, resulting in embryonic lethality, with severe cardiac structural defects, revealing the important role of *FLNA* during cardiac development [55]. In mice, the targeted ablation of *FGF13* results in embryonic lethality [56]. The *MECP2* gene is essential for embryonic development [57], acting in the differentiation of neuronal progenitors, which, in the absence of *MECP2*, do not show regular increase rates of the neuronal nuclei during development [58].

The positional and functional candidate genes associated with markers with significant recessive TRD patterns were: *TNMD*, *SLITRK4* and *BTK*. Despite the association between *TNMD* and connective and cardiovascular tissue growth and differentiation, knockout mice show normal overall morphology at birth [59]. The knockout of *Sirh7/Ldoc1* in mice exhibit an overproduction of placental progesterone and placental lactogen, resulting from abnormal placental cell differentiation and maturation, subsequently, causing delayed parturition [60].

The *TSC22D3* gene was associated with markers displaying two different TRD patterns (dam-TRD and recessive-TRD). Deficient mice for *TSC22D3* show severe testis dysplasia followed by infertility [61], reinforcing the association of this gene with male fertility-related biological processes.

### 3.10. Validation of Functional Candidate Genes Previously Identified in Independent Populations Using Genome-Wide Association Studies’ Meta-Analysis for Fertility Traits in Cattle

In addition to the previous reported results, 15 out of the 408 positional candidate genes were previously identified as functional candidate genes for cattle male fertility traits in a systematic review performed by our group [62]. These genes were: *MECP2*, *L1CAM*, *IRAK1*, *KIF4A*, *ATP6AP1*, *G6PD*, *POLA1*, *FLNA*, *IKBKG*, *HCFC1*, *EMD*, *IL13RA2*, *PAK3*, *DCX*, and *TEX11*. Some of these genes are responsible for regulating important metabolic pathways or biological processes associated with fertility. Among these processes, it is important to highlight the control of the progression of spermatogenesis, control of ciliary activity, development of Sertoli cells, DNA integrity in spermatozoa, and homeostasis of testicular cells. Additionally, the impact on fertility traits of these 15 functional and positional candidate genes was also evaluated in other species. The *MECP2* gene is associated with the development of Rett syndrome in humans [63]. In human and mice, variants located in the *MECP2* gene were previously associated with precocious puberty, undescended testes, and micropenis [64,65]. During the endometriosis development, *L1CAM* is considered a pathogenic factor in humans due to its overexpression in ovarian and endometrial carcinomas [66]. Interestingly, *IRAK1* gene is differentially expressed in impaired spermatogenesis [67]. The function of *KIF4A* transcript is crucial for the chromosome positioning and spindle formation, playing an important role during chromosome segregation in mouse oocytes [68,69]. Additionally, *KIF4A* variants were already associated with uterine capacity in beef cattle [70]. The *ATP6AP1* gene encodes the main accessory protein of V-ATPase, which is a crucial regulator of intra-organellar acidification [71]. The deficiency of *ATP6AP1* is associated with immunodeficiency, hepatopathy, and abnormal protein glycosylation in prenatal and postnatal stages [72]. In addition, *ATP6AP1* gene was described, in zebrafish, as responsible to mediate the dorsal forerunner cell proliferation and the left-right asymmetry [73]. The *GDP6* deficiency was suggested to be associated with reduced female fertility in humans [74]. The *FLNA* gene is involved with the regulation of the embryonic development of several tissues, such as the cardiac, neural and muscular tissues [55,75]. Consequently, *FLNA* gene plays a crucial role for embryo survival. The *HCFC1* gene codifies a transcriptional co-regulator responsible to control cell proliferation, which loss-of-function variants were associated with disrupted neuronal and neural progenitor cells with a deleterious effect in humans and mice [76]. In humans, copy number variants at the *PAK3* gene were found in men with Sertoli-cell-only syndrome, XY gonadal dysgenesis and premature ovarian failure [74]. The *TEX11* gene was already broadly investigated in genetic association studies regarding male fertility in cattle and other species. This gene was associated with male infertility in humans [75] and in cattle [76]. Indeed, variants in *TEX11* gene are responsible to explain up to 13% of the total genetic variance for scrotal circumference at 12 months in beef cattle [76].

## 4. Conclusions

Transmission ratio distortion analyses showing deviations from Mendelian inheritance revealed 149 SNPs displaying opposite sire-TRD between male and female offspring with strong magnitude at the beginning of the pseudoautosomal region and gradually reducing along this region until being null at the extreme of the chromosomes. This finding is evidence of the strong linkage of unique SNPs to specific-sex (Y- or X-) chromosomes and depicts the accumulation of recombination events across the pseudoautosomal region of the cattle genome. Moreover, haplotypes were found displaying recessive TRD patterns in female-offspring. Functional analyses and validation analysis using independent populations and methodologies showed that the TRD regions identified in this study were related to key biological processes and molecular functions responsible for regulating processes such as spermatogenesis, development of sertoli cells, homeostasis of endometrium tissue and embryonic development.

## Figures and Tables

**Figure 1 genes-13-02322-f001:**
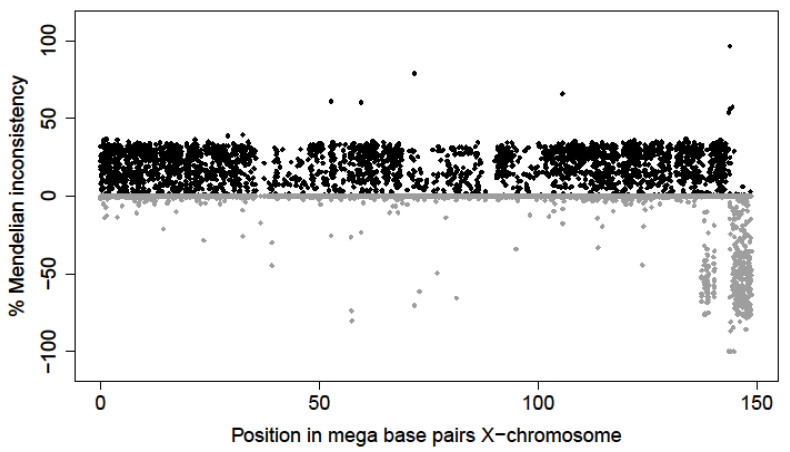
Mendelian inconsistencies across bovine X-chromosome based on the pseudoautosomal inheritance model (positive value and black color) and the heterosomal inheritance model (negative values and grey color).

**Figure 2 genes-13-02322-f002:**
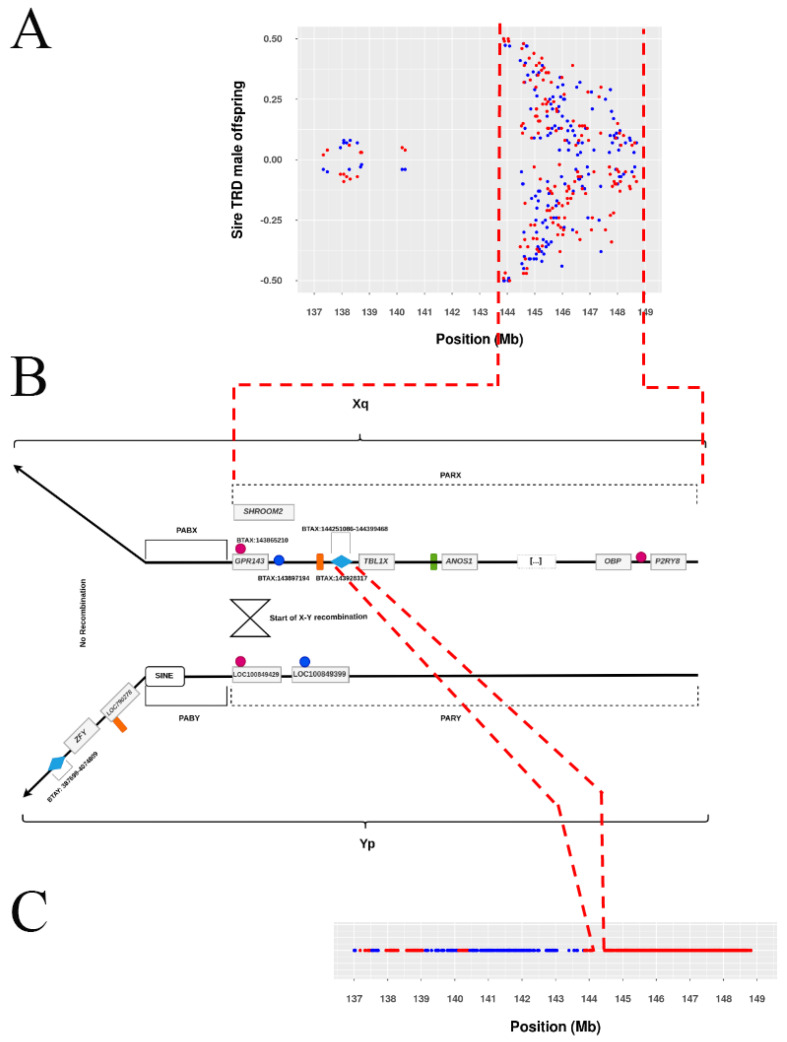
Schematic representation of the long arm of X-chromosome (Xq) and short arm of Y-chromosome (Yp) and sire-TRD patterns. (**A**) Magnitude of sire-TRD with opposite sign between male- (blue color) and female-offspring (red color) on pseudoautosomal SNPs. (**B**) Schematic representation of X and Y peudoautosomal regions (PARX and PARY) and peudoautosomal boundaires (PABX and PABY). (**C**) Density of SNP available for analyses (pseudoautosomal with red color and heterosomal with blue color). The pink and blue circles represent the first two markers with α = 0.5 (BTX:143,865,210 and BTX:143,870,595), respectively. The red circle represents the last marker with an opposite sire-TRD effect between male and female offspring (BTX:148,789,832). The orange and green rectangles correspond to the markers with opposite sire-TRD effects between male and female offspring and dam-TRD for male (orange) and female (green) offspring. The blue diamonds represent the region with highest homology between the X- and Y-chromosomes (BTX:144,251,086–144,399,458). It is important to highlight that this is a schematic image and some genes where suppressed.

**Figure 3 genes-13-02322-f003:**
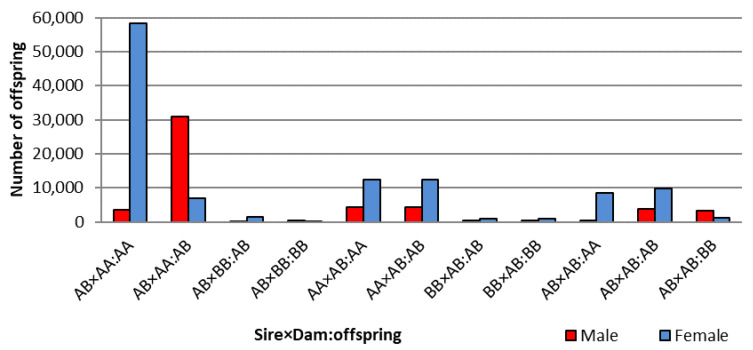
Distribution of male and female offspring for informative mattings for the SNP with highest BF with opposite sire-TRD between sex offspring.

**Figure 4 genes-13-02322-f004:**
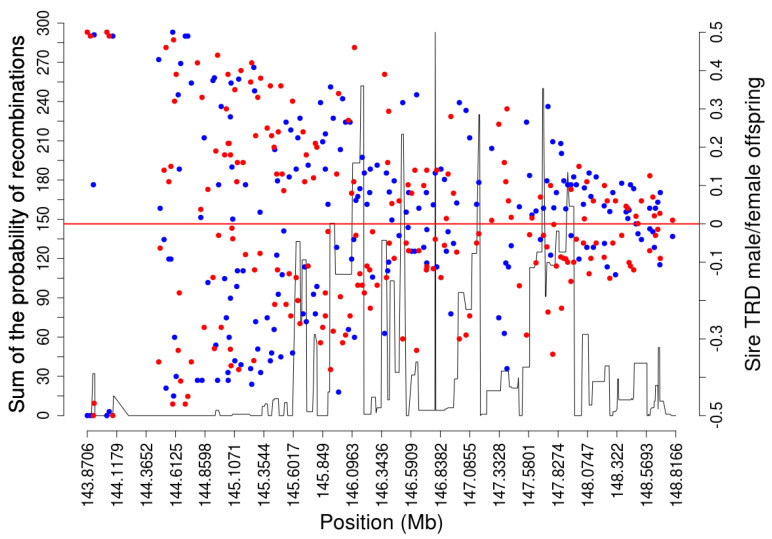
Overlapping between opposite sire-TRD (right y-axis) and sum of the probability of recombination (left y-axis) in the main pseudoautosomal part of the bovine X-chromosome (143,865,210–148,816,634 bp). The x-axis shows the genomic coordinated in Mega bases (Mb). The red and blue dots correspond to the sire-TRD magnitude in the male and female offspring, respectively.

**Figure 5 genes-13-02322-f005:**
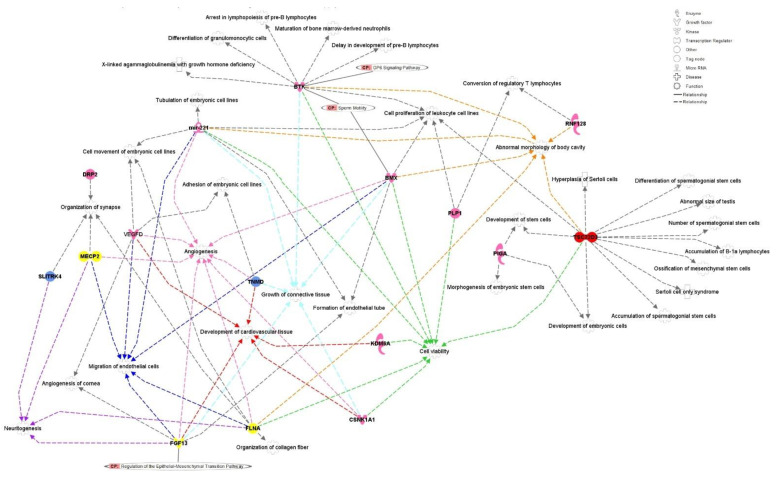
Interaction network between the positional candidate genes around TRD regions on bovine X-chromosome and biological functions and diseases identified. For the most relevant biological functions, the edges between the genes and the processes were colored in order to highlight the connection. These biological functions were: cell viability (green), angiogenesis (pink), development of cardiovascular tissue (red), migration of endothelial cells (blue), growth of connective tissue (cyan), abnormal morphology of body cavity (orange) and neuritogenesis (purple). The positional candidate genes were colored in function of the predominant (highest |α|) TRD observed for the associated marker, where: pink represents dam-TRD among in heterosomal regions (RNF128, PIGA, PLP1, BMX, KDM6A, CSNK1A1, VEGFD, DRP2 and bta-mir-221); yellow represents male- and female-dam-TRD in heterosomal regions (FLNA, FGF13 and MECP2); blue represents recessive TRD pattern in heterosomal regions (TNMD, SLITRK4 and BTK). The gene TSC22D3 was colored in red because two different TRD patterns were observed for the markers mapped in the interval used to annotate the gene. These TRD patterns were dam-TRD and recessive-TRD, both in heterosomal regions.

**Table 1 genes-13-02322-t001:** Specific sex-offspring for sire-TRD in Holstein X-chromosome.

Coordinates in Base Pairs	Frequency	Male-Offspring					Female-Offspring				
		n° Hetero ^1^ Sires	n° Hetero Dams	n° Info ^2^ Offspring	Sire-TRD (α_s♂_)	BF	n° Hetero Sires	n° Hetero Dams	n° Info Offspring	Sire-TRD (α_s♀_)	BF
137,464,119	0.63	376	2307	10,297	0.02	0.79	275	1625	7100	−0.04	10^4.2^
138,080,118	0.46	699	4368	19,985	−0.01	0.04	1281	20,752	48,445	0.03	10^22.8^
144,824,704	0.31	267	1691	8037	0.01	0.03	365	2258	8646	0.04	10^3.4^
144,927,188	0.83	7	88	228	0.36	10^3.8^	8	46	107	−0.14	0.63
145,093,201	0.24	403	2284	10,759	0.00	0.02	361	1879	9007	−0.04	10^4.2^
145,322,565	0.67	280	1532	9045	0.02	1	364	1946	9432	−0.08	10^27.4^
146,268,179	0.14	216	1351	6317	−0.15	10^58.9^	334	5575	13,299	−0.02	3.98
146,403,456	0.08	14	51	155	−0.11	0.79	13	19	115	0.29	10^5.9^
147,269,713	0.85	56	162	809	0.18	10^9.2^	33	132	442	0.01	0.08
147,433,504	0.52	302	2111	10,749	−0.07	10^18.8^	406	2793	11,668	0.02	0.63
147,587,823	0.58	98	290	1456	0.11	10^8.3^	66	209	764	−0.03	0.13
148,398,942	0.22	477	2871	14,803	0.02	10	851	14,732	37,427	0.05	10^31.7^
148,498,697	0.34	267	1685	8671	−0.04	10^2.9^	338	2173	8922	0.00	0.02
148,665,523	0.61	269	1853	8777	0.05	10^5.4^	351	2335	8873	−0.01	0.10
148,682,952	0.36	98	311	1556	−0.11	10^8.8^	67	225	795	0.03	0.13

α_s♂:_ male-sire-TRD, α_s♀:_ female-sire-TRD, BF: Bayes factor. ^1^ Number of heterozygous. ^2^ Number of informative offspring (from heterozygous parents).

**Table 2 genes-13-02322-t002:** Significant dam-TRD, male- and female-dam-TRD among heterosomal SNPs in Holstein X-chromosome.

Coordinates in Base Pairs	Offspring-Sex	n° Hetero ^1^ Dams	A × AB ^2^			B × AB			Frequency M/(F)	TRD Parameters
			AA/A ^3^	AB	B	A	AB	BB/B		
4,134,094	-	391	111	46	88	209	150	623	0.15 (0.18)	α_d_ = −0.12 (BF = 10^13.3^; ≤0.001 ^4^)
40,239,653	-	42,334	37,552	23,430	11,719	7816	15,685	22,229	0.64 (0.62)	α_d_ = 0.02 (BF = 10^20.6^; ≤0.01)
85,463,982	-	13,963	294	241	98	3401	9983	14,552	0.03 (0.04)	α_d_ = −0.02 (BF = 10^9.2^; ≤0.01)
2,230,488	M	640	854	0	668	58	0	39	0.94	α_d♂_ = 0.06 (BF = 10^4.14^; ≤0.01)
	F	477	632	628	0	0	20	18	0.93	α_d♀_ = 0 (BF = 0.06; >5)
3,873,466	M	81	88	0	56	42	0	20	0.52	α_d♂_ = 0.13 (BF = 107.2; ≤1)
	F	48	35	25	0	0	19	32	0.49	α_d♀_ = −0.01 (BF = 0.12; >5)
5,451,082	M	8389	10,984	0	10,371	1785	0	1673	0.86	α_d♂_ = 0.01 (BF = 234.4; ≤0.01)
	F	31,944	25,802	25,820	0	0	3570	3826	0.85	α_d♀_ = 0 (BF = 0.01; >5)
22,030,046	M	1364	846	0	710	931	0	795	0.46	α_d♂_ = 0.04 (BF = 10^3.2^; ≤1)
	F	1067	603	591	0	0	713	672	0.43	α_d♀_ = 0.01 (BF = 0.04; >5)
65,136,071	M	12,977	5812	0	6482	13,607	0	14,685	0.34	α_d♂_ = −0.02 (BF = 10^14.3^; ≤0.001)
	F	49,960	13,780	14,055	0	0	32,898	33,032	0.37	α_d♀_ = 0 (BF = 0.01; >5)
106,467,781	M	1627	425	0	513	1658	0	1943	0.21	α_d♂_ = −0.04 (BF = 10^4.93^; ≤1)
	F	1128	305	309	0	0	1234	1265	0.21	α_d♀_ = −0.01 (BF = 0.03; >5)
143,832,372	M	13,163	7425	0	7888	12,371	0	13,080	0.38	α_d♂_ = −0.01 (BF = 10^4.78^; ≤0.01)
	F	52,914	19,253	18,869	0	0	30,256	30,222	0.44	α_d♀_ = 0.0 (BF = 0.01; >5)
67,099,097	M	785	1192	0	1146	53	0	41	0.88	α_d♂_ = 0.01 (BF = 0.05; >5)
	F	504	936	586	0	0	28	37	0.91	α_d♀_ = 0.11 (BF = 10^14.5^; ≤0.01)
125,033,467	M	767	1116	0	1082	56	0	41	0.95	α_d♂_ = 0.01 (BF = 0.04; >5)
	F	613	880	708	0	0	45	40	0.95	α_d♀_ = 0.05 (BF = 363.1; ≤0.1)

M: male, F: female, BF: Bayes factor, α_d_: dam-TRD, α_d♂_: male-dam-TRD, α_d♀_: female-dam-TRD. ^1^ Number of heterozygous. ^2^ Sire × dam mating genotypes. ^3^ Offspring genotype. ^4^ Probability of random TRD.

**Table 3 genes-13-02322-t003:** Significant dam-TRD among heterosomal part in Holstein X-chromosome for haplotype analyses.

Coordinates in Base Pairs ^1^	n° Hetero ^2^ Dams	Dam-TRD	BF	A × AB ^3^			B × AB			Frequency %
				AA/A ^4^	AB	B	A	AB	BB/B	
134,602,363:135,936,247 ^4^	10	−0.49	10^19.80^	0	0	0	0	0	72	0.004
134,841,753:136,063,445	11	−0.49	10^19.80^	0	0	0	0	0	72	0.004
134,624,675:135,970,186	10	−0.49	10^19.51^	0	0	0	0	0	71	0.004
119,781,376:121,600,055	11	−0.44	10^19.13^	0	0	0	1	4	91	0.007
123,894,981:125,905,926	101	−0.22	10^18.12^	0	0	0	46	72	310	0.109
053,138,058:058,932,871	18	−0.38	10^16.11^	0	0	0	3	10	103	0.021
056,821,723:059,606,508	12	−0.47	10^13.65^	0	0	0	0	1	56	0.005
056,846,129:059,878,165	12	−0.47	10^13.65^	0	0	0	0	1	56	0.005
129,911,571:132,268,728	15	−0.44	10^13.11^	0	0	0	0	3	62	0.007
130,070,306:132,334,180	15	−0.44	10^13.11^	0	0	0	0	3	62	0.007
130,112,536:132,620,448	15	−0.44	10^13.11^	0	0	0	0	3	62	0.007
055,893,493:059,564,066	44	−0.24	10^12.55^	0	0	0	31	40	199	0.058
129,070,088:131,766,182	12	−0.45	10^11.64^	0	0	0	1	1	53	0.006
129,145,800:131,806,987	12	−0.45	10^11.64^	0	0	0	1	1	53	0.006
129,193,110:131,960,783	12	−0.45	10^11.64^	0	0	0	1	1	53	0.006
129,718,409:132,109,716	12	−0.45	10^11.64^	0	0	0	1	1	53	0.006
129,887,133:132,211,189	12	−0.45	10^11.64^	0	0	0	1	1	53	0.006
023,761,259:026,607,222	30	−0.29	10^11.07^	0	0	0	5	25	118	0.025
056,962,509:060,125,574	13	−0.40	10^10.59^	0	0	0	2	4	62	0.008
100,533,235:105,003,475	14	−0.38	10^10.47^	0	0	0	5	3	67	0.011

BF: Bayes factor, α_d_: dam-TRD. ^1^ Number of SNPs on the haplotype window was 20. ^2^ Number of heterozygous dams. ^3^ Sire × dam mating genotypes. ^4^ Offspring genotype.

**Table 4 genes-13-02322-t004:** Haplotype windows with recessive TRD pattern on heterosomal part of Holstein X-chromosome.

Coordinates in Base Pairs (n. SNPs ^1^)	n° Hetero ^2^ Dams	AA × AB ^3^		BB × AB		Frequency %	TRD Parameters
		AA ^4^	AB	AB	BB		
307,557: 854,184 (10)	247.00	2	8	300	281	0.40	α_g_ = −0.34 (BF = 10^3.35^); δ_g_ = 0.21; (BF = 10^3.17^)
14,897,141:16,123,681 (10)	452.00	1	8	559	579	0.60	α_g_ = −0.47 (BF = 10^12.96^); δ_g_ = 0.21; (BF = 10^6.64^)
23,895,384:27,387,980 (20)	541.00	1	7	674	634	0.90	α_g_ = −0.40 (BF = 10^12.08^); δ_g_ = 0.24; (BF = 10^10.46^)
30,116,443:30,459,667 (4)	574.00	0	6	683	689	0.90	α_g_ = −0.51 (BF = 10^19.6^); δ_g_ = 0.25; (BF = 10^11.37^)
48,750,092:52,808,614 (20)	592.00	0	5	647	669	1.10	α_g_ = −0.50 (BF = 10^17.26^); δ_g_ = 0.23; (BF = 10^8.99^)
59,564,066:61,635,072 (20)	525.00	0	8	550	549	1.80	α_g_ = −0.56 (BF = 10^19.61^); δ_g_ = 0.28; (BF = 10^11.44^)
109,041,641:110,425,524 (20)	438.00	1	8	513	539	1.20	α_g_ = −0.48 (BF = 10^12.4^); δ_g_ = 0.21; (BF = 10^5.82^)
127,431,846:129,887,133 (20)	406.00	2	9	573	551	0.80	α_g_ = −0.37 (BF = 10^8.37^); δ_g_ = 0.21; (BF = 10^6.78^)

BF: Bayes factor, α_g:_ additive-TRD, δ_g_: dominance-TRD. ^1^ Number of SNPs on the haplotype window. ^2^ Number of heterozygous dams. ^3^ Sire × dam mating genotypes. ^4^ Female-offspring genotype.

**Table 5 genes-13-02322-t005:** Top 10 canonical metabolic pathways and diseases and functions identified using the list of positional candidate genes mapped around (100 Kb up- and downstream) from candidate SNPs or within haplotype coordinates showing significant TRD.

	*p*-Value	False Discovery Rate	Genes
**Canonical Metabolic Pathway**			
Citrulline Degradation	0.0072	0.603	*OTC*
Spermine Biosynthesis	0.0145	0.603	*SMS*
PRPP Biosynthesis I	0.0282	0.603	*PRPS1*
Arginine Biosynthesis IV	0.0427	0.603	*OTC*
Proline Biosynthesis II (from Arginine)	0.0427	0.603	*OTC*
Urea Cycle	0.0427	0.603	*OTC*
Pentose Phosphate Pathway (Non-oxidative Branch)	0.0427	0.603	*TKTL1*
Glycine Cleavage Complex	0.0427	0.603	*GCSH*
Phosphatidylcholine Biosynthesis I	0.0490	0.603	*PCYT1B*
Acetyl-CoA Biosynthesis I (Pyruvate Dehydrogenase Complex)	0.0490	0.603	*PDHA1*
**Diseases and Functions**			
X-linked hereditary disease	3.21 × 10^−47^	1.6 × 10^−44^	*ADGRG2, ALG13, AMELX, AMER1, ANOS1, AP1S2, ARHGEF9, ARMCX5GPRASP2/GPRASP2, ARR3, BTK, CA5B, CDKL5, CNKSR2, COL4A6, FANCB, FGF13, FLNA, FRMD7, GLA, GPR143, HNRNPH2, IGSF1, KDM6A, KIF4A, MBTPS2, MECP2, MID2, mir221, NDUFA1, OPN1LW, OTC, PCDH19, PDHA1, PDK3, PHEX, PHKA2, PIGA, PIH1D3, PLP1, POLA1, PRPS1, RNF113A, RPS6KA3, RS1, SERPINA7, SMPX, SMS, SRPX2, TIMM8A, TSPAN7, UBE2A, ZC4H2, ZNF674*
X-linked mental retardation	1.47 × 10^−15^	3.7 × 10^−13^	*AP1S2, CNKSR2, HNRNPH2, KDM6A, KIF4A, MECP2, MID2, PDHA1, RNF113A, RPS6KA3, SMS, SRPX2, TSPAN7, UBE2A, ZC4H2, ZNF674*
X-linked hearing loss	1.29 × 10^−8^	2.2 × 10^−6^	*ARMCX5GPRASP2/GPRASP2, COL4A6, PRPS1, SMPX*
Syndromic X-linked mental retardation	1.97 × 10^−8^	2.5 × 10^−6^	*AP1S2, CNKSR2, HNRNPH2, KDM6A, MECP2, SMS, SRPX2, UBE2A, ZC4H2*
Familial mental retardation	4.28 × 10^−8^	4.3 × 10^−6^	*AP1S2, CNKSR2, FLNA, HNRNPH2, KDM6A, KIF4A, MECP2, MID2, PDHA1, RNF113A, RPS6KA3, SMS, SRPX2, TSPAN7, UBE2A, ZC4H2, ZNF674*
X-linked nonsyndromic sensorineural hearing loss	1.46 × 10^−6^	1.2 × 10^−4^	*COL4A6, PRPS1, SMPX*
Autism or intellectual disability	1.95 × 10^−6^	1.4 × 10^−4^	*AP1S2, CDKL5, CNKSR2, FLNA, HNRNPH2, KDM6A, KIF4A, MECP2, MID2, PDHA1, RNF113A, RPS6KA3, SMS, SRPX2, TSPAN7, UBE2A, ZC4H2, ZNF674*
Familial syndromic intellectual disability	2.57 × 10^−6^	1.6 × 10^−4^	*AP1S2, CNKSR2, FLNA, HNRNPH2, KDM6A, MECP2, SMS, SRPX2, UBE2A, ZC4H2*
Childhood onset epilepsy syndrome	3.63 × 10^−6^	2.0 × 10^−4^	*ARHGEF9, CDKL5, SRPX2*
Cognitive impairment	3.49 × 10^−5^	1.7 × 10^−3^	*AP1S2, ARHGEF9, CA5B, CNKSR2, FLNA, HNRNPH2, KDM6A, KIF4A, MECP2, MID2, PDHA1, RNF113A, RPS6KA3, SMS, SRPX2, TSPAN7, UBE2A, VEGFD, ZC4H2, ZNF674*

## Data Availability

Data that support the findings of this study were provided by the Canadian Dairy Network, a member of Lactanet (Guelph, ON, Canada), which were used under a material transfer agreement and thus are not publicly available.

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
