# Peer review of "Deviations from Mendelian Inheritance on Bovine X-Chromosome Revealing Recombination, Sex-of-Offspring Effects and Fertility-Related Candidate Genes"

_genes, 2022, doi:10.3390/genes13122322_

Round 1

Reviewer 1 Report

The current manuscript investigates “Deviations from Mendelian inheritance on bovine X-chromosome revealing recombination, sex-of-offspring effects and fertility-related candidate genes”. The paper is well-written and comprehensible. The topic is interest, the all approaches were well conducted, and the manuscript also raised many concerns. The follow are some specific comments.

·         Please add a brief summary of the results of previous research on this topic in the introduction

·         Please separate the results from the discussion

·         Please summarize the conclusion and focus on the most important results obtained

Author Response

The current manuscript investigates “Deviations from Mendelian inheritance on bovine X-chromosome revealing recombination, sex-of-offspring effects and fertility-related candidate genes”. The paper is well-written and comprehensible. The topic is interest, the all approaches were well conducted, and the manuscript also raised many concerns. The follow are some specific comments.

Please add a brief summary of the results of previous research on this topic in the introduction

Answer: Thank you for your suggestion. In the updated version of the manuscript it was included the information of previous studies reporting TRD results in other species, as well as the consequences of TRD on genetics studies (Lines 37-42).

Please separate the results from the discussion

Answer: Thank for your suggestion. However, the guidelines from Genes allows the presentation of results and discussion merged in a single section. The number of analysis and results available in the current study made us to choose the presentation of each result followed by the correspondent discussion in order to make it easier to follow. Therefore, we decided to keep this writing style as it is under the guidelines of manuscript formatting from Genes.

Please summarize the conclusion and focus on the most important results obtained

Answer: Thank you for your suggestion. The conclusion was summarized in the updated version of the manuscript focusing on the most important results obtained (Lines 570-581).

Reviewer 2 Report

This study uncovered the prevalence of different Transmission Ratio Distortion (TRD) patterns across both heterosomal and pseudoautosomal regions of the X-chromosome and revealed functional candidate genes for bovine reproduction. These genes are potential molecular markers and may be used in further research. The manuscript is well written, the figure; figure legends and tables are properly presented.

Author Response

Review 2

This study uncovered the prevalence of different Transmission Ratio Distortion (TRD) patterns across both heterosomal and pseudoautosomal regions of the X-chromosome and revealed functional candidate genes for bovine reproduction. These genes are potential molecular markers and may be used in further research. The manuscript is well written, the figure; figure legends and tables are properly presented.

Answer: Thank you for your comments.
